# Effects of Different Phospholipid Sources on Growth and Gill Health in Atlantic Salmon in Freshwater Pre-Transfer Phase

**DOI:** 10.3390/ani13050835

**Published:** 2023-02-24

**Authors:** Renate Kvingedal, Jannicke Vigen, Dominic Nanton, Kari Ruohonen, Kiranpreet Kaur

**Affiliations:** 1Cargill Aqua Nutrition, 4335 Dirdal, Norway; 2Aker BioMarine Antarctic ASA, 1366 Lysaker, Norway

**Keywords:** krill, phospholipid, salmon, smolt, feed, gills

## Abstract

**Simple Summary:**

Optimal nutrition is important for Norwegian-farmed Atlantic salmon in the challenging early seawater phase, which shows a higher mortality leading to significant economic losses. Phospholipids are reported to enhance growth, survival, and health in the early stages of the fish life. Atlantic salmon (74 to 158 g) were fed six test diets to evaluate alternative phospholipid (PL) sources in freshwater and were transferred to a common seawater tank with crowding stress after being fed the same commercial diet up to 787 g. Krill meal (KM) was evaluated using dose response with the highest 12% KM diet compared against 2.7% fluid soy lecithin and 4.2% marine PL (from fishmeal) diets, which were formulated to provide the same level of added 1.3% PL in the diet similar to base diets with 10% fishmeal in the freshwater period. A trend showing increased weight gain with high variability was associated with an increased KM dose in the freshwater period but not during the whole trial, whereas the 2.7% soy lecithin diet tended to decrease growth during the whole trial. No major differences were observed in liver histology between the salmon that were fed different PL sources during transfer. However, a minor positive trend in gill health based on two gill histology parameters was associated with the 12% KM and control diets versus the soy lecithin and marine PL diets during transfer.

**Abstract:**

Growth and histological parameters were evaluated in Atlantic salmon (74 g) that were fed alternative phospholipid (PL) sources in freshwater (FW) up to 158 g and were transferred to a common seawater (SW) tank with crowding stress after being fed the same commercial diet up to 787 g. There were six test diets in the FW phase: three diets with different doses of krill meal (4%, 8%, and 12%), a diet with soy lecithin, a diet with marine PL (from fishmeal), and a control diet. The fish were fed a common commercial feed in the SW phase. The 12% KM diet was compared against the 2.7% fluid soy lecithin and 4.2% marine PL diets, which were formulated to provide the same level of added 1.3% PL in the diet similar to base diets with 10% fishmeal in the FW period. A trend for increased weight gain with high variability was associated with an increased KM dose in the FW period but not during the whole trial, whereas the 2.7% soy lecithin diet tended to decrease growth during the whole trial. A trend for decreased hepatosomatic index (HSI) was associated with an increased KM dose during transfer but not during the whole trial. The soy lecithin and marine PL diets showed similar HSI in relation to the control diet during the whole trial. No major differences were observed in liver histology between the control, 12% KM, soy lecithin, and marine PL diets during transfer. However, a minor positive trend in gill health (lamella inflammation and hyperplasia histology scores) was associated with the 12% KM and control diets versus the soy lecithin and marine PL diets during transfer.

## 1. Introduction

Farmed salmon are typically transferred from early phase production in tanks on land to seawater cages that constitutes a challenging environment, where fish can experience significant mortality before reaching harvest size. For example, mortality in Atlantic salmon ranged from 15 to 16% from 2017 to 2021 in Norway, with approximately 35% of sea cage mortality occurring in the first 0–3 months at sea for the 2010–11 salmon generations in the Norwegian-farmed Atlantic salmon [1]. This mortality in the early sea cage phase leads to significant economic loss [2]. Thus, research on optimal nutrition to produce robust smolts for improved survival and growth after transfer to the sea cage is of interest to the aquaculture industry [3]. Fish meal (FM) and fish oil (FO) dominated early commercial salmon feed formulations and provided essential nutrients, but usage of these marine ingredients has declined over time as they are limited resources at generally higher prices compared to alternative ingredients where sustainability measures are also considered [4]. Antarctic krill meal (KM; Euphausia superba) is a commercially known ingredient in salmon feeds, with potential benefits toward enhancing growth and health in salmonids [5]. The krill fishery in the Antarctic Southern Ocean is considered highly regulated and sustainable [6,7]. KM provides a range of nutrients including proteins (similar amino acid profile to FM); water soluble nitrogenous components (free amino acids, peptides, nucleotides, and trimethylamine N-oxide), which can act as potential feed attractants; astaxanthin; marine omega-3 fatty acids (eicosapentaenoic acid (EPA) and docosahexaenoic acid (DHA)); and phospholipids (PLs) [5]. Substantial evidence exists showing that dietary PL can improve growth, survival, and health (reduced intestinal steatosis and deformities) in the larval and early juvenile stages of the fish [8,9,10,11]. In addition, KM and krill oil (KO) reduced fat accumulation in the hepatocytes in comparison to soybean lecithin as the PL sources in the diet of seabream larvae [10,12,13]. In addition, there was an indication that seabream juveniles that were fed a diet with 9% KM had lower hepatocyte vacuolization (fat storage) versus a control diet without KM that was higher in fishmeal [12,13], and a non-significant trend for lower hepatocyte vacuolization was indicated for seabream larvae that were fed a diet with krill oil versus soybean lecithin as the PL source [10]. PLs from different sources can have different properties. KM has approximately 40% PL consisting of the total lipid with phosphatidylcholine (PC) at >80% of the total PL and ca. 18% EPA + DHA of the total lipid [14]. In comparison, fluid soy lecithin can have approximately 46% PL of product (does not include glycolipids and complex sugars) with ca. 35% PC of the total PL and ca. 55% 18:2n-6 of the total FA as the major FA with no EPA + DHA [15]. KM has been documented in the diet of seawater salmon [16,17,18], however, only KO has been documented in the diet of freshwater salmon during the pre-transfer to the seawater phase [19]. The objective of the present study was to document the effect of the KM dose as a source of PL and compare it against other PL sources in the feed of freshwater Atlantic salmon during the pre-transfer phase followed by the early seawater phase by evaluating the growth and histological health parameters. A four-level graded dose response for KM up to 12% of the diet along with a comparison of alternative PL sources (soy lecithin and marine PL from fishmeal) formulated to provide the same level of added 1.3% PL in the diet as 12% KM was evaluated in freshwater diets for salmon during the pre-transfer phase. Fish identified by pit tag with this pre-transfer freshwater feeding history were then transferred to a common seawater tank with crowding stress after transfer and a drop in water temperature at transfer (crowding and water temperature drop can be experienced at transfer commercially) and then were fed the same commercial feed. Gill and liver histology were also compared for salmon that were fed the alternative PL source diets at the end of the freshwater pre-transfer period.

## 2. Materials and Methods

### 2.1. Feed Formulation and Composition

Three different sources of PL were tested in pre-transfer freshwater feeds: (i) krill meal (Qrill^TM^ Aqua; Aker BioMarine Antarctic ASA) at four levels for dose response (4%, 8%, and 12% of diet), (ii) fluid soy lecithin as a vegetable PL source, and (iii) marine phospholipid-rich oil sourced from North Atlantic fish species from Triple 9 (TripleNine, Trafikhavnskaj 9, DK-6700 Esbjerg, Denmark)). , and a control diet. The trial diets are referred to as Control, KM4, KM8, KM12, VegPL, and MarPL, respectively. Trial feeds were formulated using a commercial formulation program with external oil mix calculations and produced by extrusion at Cargill Innovation Center (Dirdal, Norway) for ca. 74 g fish with lipid nutrients and then adjusted for purposes of the trial. The 4-mm pre-transfer freshwater trial feeds were formulated and analyzed to have similar digestible energy (22.1–23.6 MJ/kg gross energy), protein (46–49% range), and fat (22–24% range) (Table 1) and with similar calculated 1.1% EPA + DHA in diet, 15–16% saturated in total FA, and 1.3 n-6/n-3 fatty acid (FA) ratio across trial feeds. Protein was analyzed by the Dumas principle using the Elementar Rapid Max N system. Fat was analyzed by low-field nuclear magnetic resonance scan using the NMR Analyzer Bruker minispec mq10 system (Cargill Innovation Center, Dirdal, Norway). Gross energy was analyzed by the Leco gross energy bomb calorimetry system (Cargill Innovation Center, Dirdal, Norway). Moisture was predicted by the NIR FOSS DS2500 system (Cargill Innovation Center, Dirdal, Norway) by using the feed model at Cargill. A similar 1.3% PL in diet across pre-transfer freshwater diets was calculated from the addition of 12% krill meal, fluid soy lecithin, and marine PL test ingredients to base formulations with the same 10% fishmeal level across the diets. There was variation in the other ingredients (added oil, plant ingredients, and micronutrients) needed for balancing or reaching nutrient targets. The choline level was formulated to be the same for control and VegPL diet with MarPL and KM12 providing additional choline to these diets in the form of phosphatidylcholine (PC). However, formulated choline levels for control diet and fluid soy lecithin diets were in excess of the NRC 2011 requirements for salmonids and in excess of the lowest choline level used by Hansen and coworkers [20] with no growth differences observed (1340 to 4020 mg choline/kg diet dose response trial for 456 g initial weight salmon). Lipid accumulation in the gut was reduced for salmon (456 g initial weight) at increased choline levels [20]. The formulation and composition of feeds are given in Table 1.

### 2.2. Fish Trial Conditions

The experiment was performed according to the guidelines and protocols approved by the European Union (EU Council 86/609; D.L. 27.01.1992, no. 116) and by the National Guidelines for Animal Care and Welfare published by the Norwegian Ministry of Education and Research.

*Atlantic salmon* (*Salmo salar*) with an initial weight of ca. 67 g were used for the trial. The fish were pit-tagged and randomly distributed into 24 freshwater flow-through tanks (1 m diameter and 0.45 m^3^ volume) to have 40 fish per tank at the start of trial diet feeding. These fish after 15 days of tank acclimation were 74 ± 12 g (average ± SD for all 960 fish in 24 tanks at the start of trial feeding) and then were fed the freshwater pre-transfer trial diets (Table 1) over a 53-day period. Water temperature averaged 14.3 °C (13.3–15.3 °C range) with 107% average oxygen saturation at the inlet and 90% oxygen saturation at the outlet during the freshwater acclimation and trial diet feeding period. Fish were fed the six trial diets to four replicate tanks during the 53-day freshwater pre-transfer period using an automatic belt feeder with continuous feeding for 20 h per day in excess of satiation level. Feed intake was calculated on a weekly basis by collecting and weighing uneaten pellets as well as by weighing the amount fed. There was a 12 h light: 12 h dark photoperiod regime from Day 0 at freshwater tank acclimation to Day 33 after which a 24 h light regime was used to initiate smoltification. After this freshwater pre-transfer feeding period, fish from all the tanks (17–20 fish per tank from the 24 freshwater tanks) were transferred to a larger common seawater flow-through tank (5 m diameter and 21.6 m^3^ volume with 28.5 ppt salinity, and no acclimation time from 0 ppt freshwater to 28.5 ppt seawater) with a water temperature drop at transfer (ca. 14 to 9 °C) and crowding stress after transfer (lowered water level to ca. 20 cm for one hour with supplemental oxygen for all 459 fish of ca. 167 g within a ca. 0 to 30 h period after transfer) in the common seawater tank after all 17–20 fish per tank from the 24 freshwater tanks fish were transferred over and then were fed a common commercial extruded salmon diet (EWOS AS) for a further 98 days. Daily water temperature was lower during the seawater phase averaging 9.4 °C (8.5–11.1 °C range).

### 2.3. Fish Growth

The 40 fish per tank were weighed individually with pit-tag identification on acclimation to the freshwater tanks (Day 0), at the start of trial diet feeding (Day 15), at intermediate weighing (Day 33), and after 53 days of trial feeding in the freshwater (Day 68). The fish weight gain in the freshwater pre-transfer period from Day 15 (start of freshwater trial diet feeding) to 68 were compared statistically between diets. A total of 17–20 fish from each of the 24 freshwater tanks were transferred to the common seawater tank on Day 68 with fish weighing performed on Days 35, 73, and 98 after transfer to seawater. There were 9 to 17 fish representing the original tanks in the freshwater period with 50 to 58 fish representing each of the test diets from the freshwater period at final weighing in seawater at 98 days after transfer to the common seawater tank. The fish weight gain over the whole trial period in freshwater and seawater from Day 15 to 166 days were statistically compared between diets.

### 2.4. Hepatosomatic Index

Hepatosomatic index (HSI) is the liver weight percent of the whole body weight. HSI was measured on 10 fish randomly sampled per tank (four tank replicates per diet) to study 40 fish per diet at the end of the freshwater pre-transfer period when fed test diets and 40 fish per diet (identified by pit-tag) at the end of the seawater phase when fed the common commercial diet.

### 2.5. Histology

Gill and liver histology were performed on the fish involved in the dietary phospholipid source comparison (KM12, VegPL, and MarPL) and on fish fed the Control diet at the end of the freshwater pre-transfer period. Liver (half tissue section) and gill (left gill arch 2) tissues were randomly sampled from five fish per tank to give a total of 20 liver and 20 gill tissues per diet group for histological analysis. The tissues were fixed in formalin (4% formaldehyde) and stored at room temperature until sent to Pharmaq Analytiq AS (Harbitzallée 2A, 0275 Oslo, Norway) for histological analysis.

### 2.6. Statistical Analysis

The weight gain for the different periods was modelled by computing the weight gain of each tagged individual and then using a hierarchical generalized additive model (GAM) with the spline function to describe the possibly non-linear dose-response. A random effect of tank was added to the model to account for the multiple individual observations per experimental unit. The total feed intake over the periods of interest was modelled with a single level GAM with a spline function describing the dose-response function. Hepatosomatic index (HSI) was modelled by a hierarchical GAM model using a spline function to describe the dose-response function, mean-centered round weight of the fish as a covariate, and a random effect of tank to account for the multiple individual observations per tank. From this model the expected liver weight was solved for an average-sized sampled fish and expressed as HSI by dividing the expected liver weight with the mean round weight of the sample. Gill and liver histology scores are ordinal variables for which common arithmetic operations, such as sum or mean, are not defined and therefore scores require an ordinal model returning the score probability for evaluation. A hierarchical GAM for ordinal data was set up by using a spline function to describe the dose-response function, and a random effect of tank was included to account for multiple individuals observed per tank. The models for weight gain, feed intake, and HSI assumed the error distribution is the normal distribution, and the model for gill and liver scores assumed the model is ordinal and the errors followed the ordered categorical family. All data processing and statistical modelling was conducted with the R language [21]. The GAMs were estimated with the “gam” function of the R language add-on package “mgcv” [22].

The outcomes from the fitted statistical models are presented graphically by showing the mean response and the 95% credible intervals. The mean (median) response and the 95% credible intervals were computed with the help of a parametric bootstrap (with 10,000 random draws per parameter) by taking the 25%, 50%, and 97.5% quantiles of the computed response vector. In the case of a categorical predictor variable (for comparing the different PL sources), the graphs show the mean and an error bar of the 95% credible interval. In the case of a continuous predictor (for the dose-response of krill meal inclusion), the mean response is shown as a median dose-response curve and the 95% credible interval is shown as a confidence band around the mean curve. This way both the magnitude of any potential effect (biological significance) and the uncertainty of any effect estimate (statistical significance) can be shown in the same graph for all the results independent of the response following the normal, binomial, or ordered categorical distribution.

## 3. Results

### 3.1. Growth Performance

Atlantic salmon of 74 g (overall tank average) were fed the six test diets up to 158 g (overall tank average), growing 2.1-times the initial fish weight to the end of the freshwater pre-transfer period. There was no clear trend for increased feed intake with KM dose in the FW pre-transfer phase (Figure 1). A trend for increased feed intake was indicated for the Control and KM12 diets compared to the MarPL and VegPL diets in the PL source comparison for the FW pre-transfer phase (Figure 2). There was overall high variability for the feed intake comparisons. A trend for increased fish weight gain with high variability was indicated with increased KM dose in the FW phase (Figure 3). There was similar weight gain during the whole trial with feeding the KM dose in the FW pre-transfer phase followed by feeding the same commercial diet in a common tank for the SW phase (Figure 4). Fish fed the KM12 diet had increased weight gain compared to the VegPL diet with the MarPL and Control diets having intermediate weight gains in the PL source comparison for the FW pre-transfer phase (Figure 5). Weight gain was similar for the fish that were fed KM12, MarPL, and Control diets, with a trend for higher indicated weight gain than the VegPL group during the whole trial, with feeding the KM dose in the FW pre-transfer phase followed by feeding the same commercial diet in a common tank for the SW phase (Figure 6, Appendix A).

### 3.2. Hepatosomatic Index

A trend for decreased hepatosomatic index (HSI; liver% of fish weight) was indicated for the fish that were fed increased KM dose from 0 to 12% of diet at the end of the freshwater pre-transfer feeding phase (Figure 7). There was no decrease in HSI with feeding KM dose at the end of the whole trial after the FW pre-transfer phase followed by feeding the same commercial diet in a common tank for the SW phase (Figure 8). A lower HSI was indicated for the fish that were fed the KM12 diet compared with the fish that were fed the MarPL, VegPL, and Control diets at the end of the freshwater pre-transfer feeding phase (Figure 9) with a similar minor HSI trend observed over the whole trial (Figure 10).

### 3.3. Histology

#### 3.3.1. Gill Histology

An increased probability for very mild to mild gill lamella inflammation and hyperplasia score was indicated for the salmon that were fed the VegPL and MarPL diets compared to the Control and 12% KM diets at the end of the freshwater pre-transfer phase after 53 d of feeding the trial diets (Figure 11a,b). Other following gill histology responses were evaluated with no major differences between the diets: vascular lesions, filament inflammation, necrosis of respiratory epithelium, necrosis affecting deeper tissues, fusion of lamella ,and other lesions noted as present or absent.

#### 3.3.2. Liver Histology

No major differences were observed in liver histology between the control, 12% KM, soy lecithin, and marine PL diets at the end of the FW pre-transfer phase after 53 d of feeding the trial diets (data not shown). The following liver histology responses were evaluated: total amount of abnormal tissue, inflammation, necrosis, inflammation in liver tissue or capsule (peritonitis), peribiliary or perivascular inflammation, neoplasia, fibrosis, lipid deposition, other degenerative changes, vascular lesions, and other lesions noted as absent or present.

## 4. Discussion

The present study evaluated the effect of different phospholipid sources fed over 53 d in the freshwater pre-transfer phase followed by feeding the same commercial diet over 98 d in a common seawater tank on growth performance and health parameters of Atlantic salmon. KM was evaluated in dose response (4%, 8%, and 12.0% of diet), and diets with 2.7% fluid soy lecithin (VegPL) and 4.2% MarPL as alternative PL sources were formulated to provide the same level of added 1.3% PL in diet as 12% KM. All the test diets contained 10% fishmeal in the FW phase. A trend was indicated for increased fish weight gain (high variability) with increased KM dose in the FW pre-transfer phase but a carry-over effect on growth was not observed for the same salmon fed the same commercial diet after seawater transfer. Salmon (104 g initial weight) that were fed krill meal at 7.5 and 15% of diet for higher fishmeal diets (40–52% of diet range) than the current trial had increased growth after transfer to sea cage [16]. Fishmeal provides PL, so higher fishmeal diets may reduce the need for KM as a PL source [23]. However, KM also provides amino acids (protein), water-soluble nitrogenous components (potential feed attractants), astaxanthin, and EPA + DHA, hence, it is more than a PL source. KM feeding may need to continue after sea water transfer to have a positive effect on growth at the end of the trial, noting the positive effects of KM on salmon growth observed in other but not all trials, which can depend on life stage and challenges, diet composition, KM refining (de-shelling etc.), and inclusion level [5].

A trend for decreased fish weight gain was indicated for the VegPL diet in the FW phase and over the whole trial compared with the control diet, whereas the MarPL diet showed more similar growth to the control diet over the whole trial, noting that only one PL level tested for MarPL and fluid soy lecithin matched that provided by KM12, so optimal dose was not evaluated. The choline level was formulated to be the same for the control and VegPL diets with KM12 and MarPL providing additional choline to these diets in the form of phosphatidylcholine (PC). Formulated choline levels for the control diet and fluid VegPL diets were in excess of the NRC 2011 requirements for salmonids and in excess of the lowest choline level used by Hansen et al. in 2020 with no growth differences observed (1340 to 4020 mg choline/kg diet dose response trial for 456g initial weight salmon) [20]. Lipid accumulation in the gut was reduced for these salmon (456 g initial weight) at increased choline levels [20]. Effects of increased choline with KM inclusion cannot be ruled out and further research would be needed to separate choline from PL effects for these smaller pre-transfer salmon (74 to 158 g fish weight) that were fed lower fat pre-transfer diets (22–24% fat) than during the seawater growth with choline requirements for reducing the lipid accumulation in the intestine, potentially dependent on dietary fat level [20]. Higher growth was generally observed for PL provided by KO over soy lecithin at various PL doses for the first-feeding stage of salmon, but this growth trend was not consistent at various PL doses over the whole trial from the first-feeding to smolt [19]. PL from KO was indicated to be more effective than fluid soy lecithin for reducing intestinal steatosis in smaller salmon (2.5 g salmon, but no steatosis observed across diets for 10–20 g salmon) and low level of vertebral deformities [19]. Marine PL sources (FM and KO) were also compared against soy lecithin at a similar ca. 3.5% PL of diet level for the first-feeding Atlantic salmon (0.14 g initial weight) with these PL sources, giving similar growth to ca. 2.4 g final fish weight with no conclusive mortality or intestinal histology differences between PL sources but these parameters were generally improved for the PL source diets with higher PL compared to the control diets with lower PL. An uncertain observation of higher average growth was indicated for the marine PL sources over soy lecithin at intermediate weighing for salmon at ca. 0.6 g [24]. Effects of PL cannot be isolated from KM but the increased growth for KM12 over the VegPL diet in the pre-transfer phase may be due to PL, choline, water soluble nitrogenous components, etc., noting that there was also an indicated trend for decreased growth of VegPL versus the control diet in the pre-transfer phase.

Addition of KM did not give a clear increase in feed intake compared to the control diet and there was an indicated trend of decreased feed intake for the MarPL and VegPL diets, but strong conclusions cannot be made due to the high variability. Feed intake can only be measured on a tank basis, so it was not possible to estimate feed intake of fish with different pre-transfer freshwater feeding histories in a common tank that were fed the same diet in the seawater phase.

A trend for decreased hepatosomatic index (HSI) was indicated with increased KM inclusion and for the 12% KM diet versus the other PL sources added to provide the same PL level in the pre-transfer phase, but the effect of KM on decreasing HSI was not carried over into the seawater phase with fish that were fed the same diet in a common tank (Figure 7, Figure 8, Figure 9 and Figure 10). There was no difference in the liver lipid droplet accumulation based on histology (normal scores only) for salmon that were fed the diets containing different PL sources at the end of the freshwater pre-transfer period. The lower HSI in KM12 could be due to the positive effects from krill PL (and choline) on the lipid transport and deposition in organs, with this effect of feeding 12% KM to Atlantic salmon documented by [17] with less pale livers and reduced liver fat. The authors further supported this observation with a significantly higher expression of the cadherin 13 (Chd) gene in the 12% KM group associated with circulating levels of the adipocyte-secreted protein adiponectin that has potential anti-inflammatory effects and plays an important role in metabolic regulation and is associated with the fatty liver index in humans [25]. However, Chd expression was not studied in the current study, and hence, further studies are warranted to explore the association between Chd expression, his, and absolute fat accumulation in the liver in salmon. Increased choline, which KM provided in this trial, was shown to reduce fat accumulation in the intestine of Atlantic salmon [20]. Choline supplementation was also indicated to reduce HSI in Atlantic salmon, but this was not reflected in lower liver fat or histological vacuolization, noting that there are variable trends of dietary choline deficiency on the liver fat level of fish reported in the literature [26]. PL from KO was indicated to be more effective than fluid soy lecithin for reducing intestinal steatosis in smaller salmon (2.5 g salmon but no steatosis observed across diets for 10–20 g salmon). Further studies are required to associate higher liver fat with welfare in salmon.

Gills are one of the most vital organs of fish, due to their function in respiration, osmoregulation, excretion of nitrogenous waste, pH regulation, and hormone production [27]. Gill health has become one of the most significant health and welfare challenges in the salmon aquaculture industry in Norway, Scotland, and Ireland [28,29,30]. The gill disorders are generally complex and multifactorial and are related to both biological factors, such as parasites and pathogens, handling stress, treatments, or due to the environmental factors, such as temperature, salinity, algal blooms, etc. Hence, the gill diseases are challenging to prevent and control and lead to high mortality, reduced production performance, and impaired fish welfare, cumulating in huge economic losses [31]. There were no differences reported for histological parameters investigated except in the presence of ectopic epithelial cells containing mucus in the lamina propria in the hindgut (potential inflammatory marker) of salmon (grown from 2.3 to 3.9 kg in sea cages) that were fed 15% fishmeal diet but not for 12% KM of diet in a 5% fishmeal diet, which may suggest anti-inflammatory effects of KM [17]. KM provides astaxanthin (166 mg/kg in the KM used for the present study) to the diet as a natural antioxidant with potential anti-inflammatory properties [32]. KM and MarPL also provide EPA + DHA attached to PL, which may affect bioavailability of EPA + DHA for use in cell membranes and inflammatory response [33] but this is not documented in fish. In the current study, there was decreased probability for very mild to mild gill lamella inflammation and hyperplasia scores indicated in salmon that were fed 12% KM compared to the soy lecithin and marine PL diets but gill histology for salmon that were fed the 12% KM diet was similar to the control diet without KM (Figure 5).

## 5. Conclusions

Overall, increased KM tended to increase growth (high variability), whereas the VegPL diet tended to decrease growth compared to the control diet in the FW pre-transfer phase. The positive growth trend indicated for KM fed pre-transfer was not carried over into the seawater phase for fish fed the same diet. A minor positive trend in gill health (lamella inflammation and hyperplasia histology scores) was indicated for the 12% KM and Control diets compared with the VegPL and MarPL diets in the FW pre-transfer phase. Hepatosomatic index tended to decrease with KM fed in the pre-transfer phase, noting that all livers evaluated by histology were considered normal for lipid droplet accumulation. Only one VegPL and MarPL dose was tested, so dose effect of these PL sources and comparison with krill oil to better isolate the PL effect from other nutrients in KM as well as a post-transfer feeding comparison of these PL sources could be areas to research further in transfer diets for salmon.

## Figures and Tables

**Figure 1 animals-13-00835-f001:**
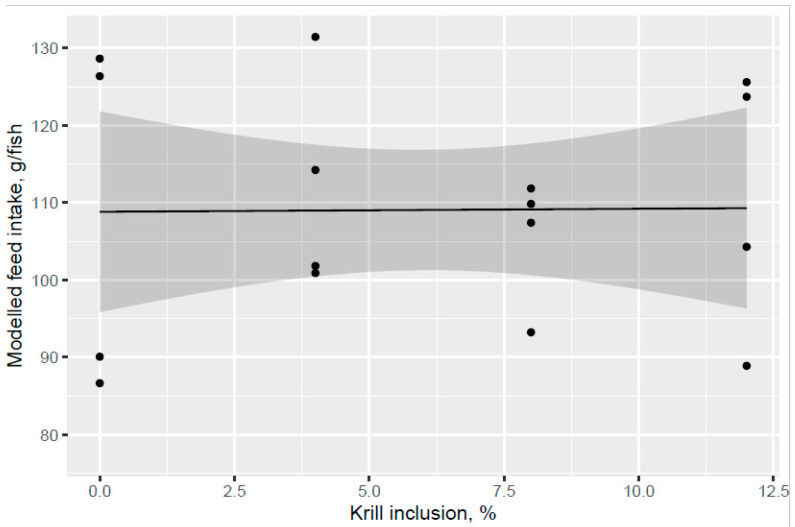
Modelled feed intake in relation to increasing KM inclusion after 53 d of feeding in the freshwater pre-transfer phase. Dots denote observed tank mean feed intake with error bars giving 95% credible intervals.

**Figure 2 animals-13-00835-f002:**
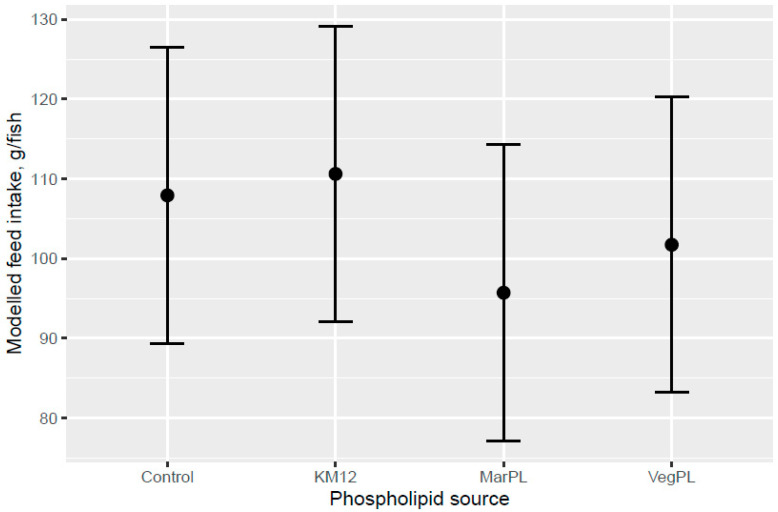
Modelled feed intake for diets containing different phospholipid sources after 53 d of feeding in the freshwater pre-transfer phase. Dots denote observed tank mean feed intake with error bars giving 95% credible intervals.

**Figure 3 animals-13-00835-f003:**
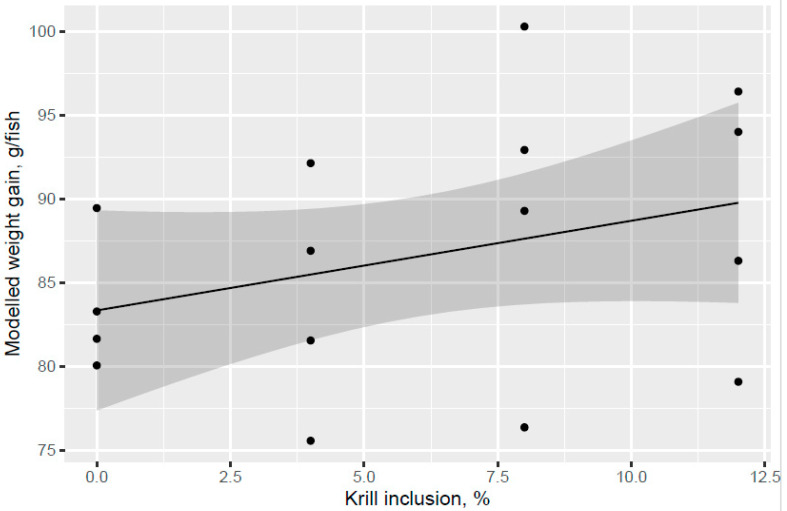
Modelled weight gain with increasing KM inclusion after 53 d of feeding in the freshwater pre-transfer phase plotted with 95% credible intervals as the grey band. Dots denote observed tank mean weight gain.

**Figure 4 animals-13-00835-f004:**
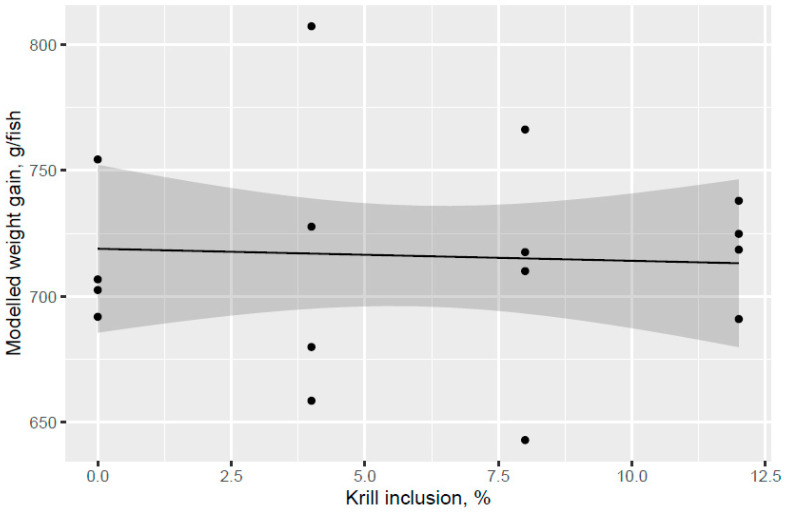
Modelled weight gain with increasing KM inclusion over 53 d in the freshwater pre-transfer phase followed by feeding the same commercial diet over 98 d in the common seawater tank plotted with 95% credible intervals as the grey band. Dots denote observed tank mean weight gains.

**Figure 5 animals-13-00835-f005:**
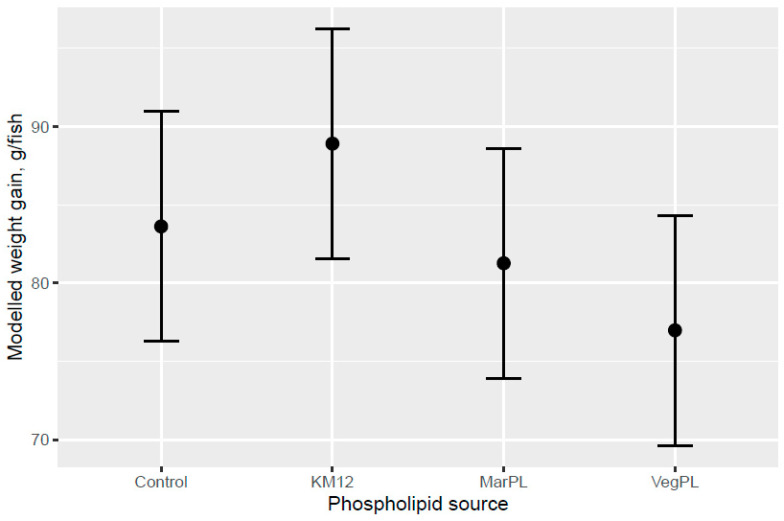
Modelled weight gain for feeding diets with different phospholipid sources after 53 d of feeding in the freshwater pre-transfer phase plotted with 95% credible intervals. Dots denote observed tank mean weight gains.

**Figure 6 animals-13-00835-f006:**
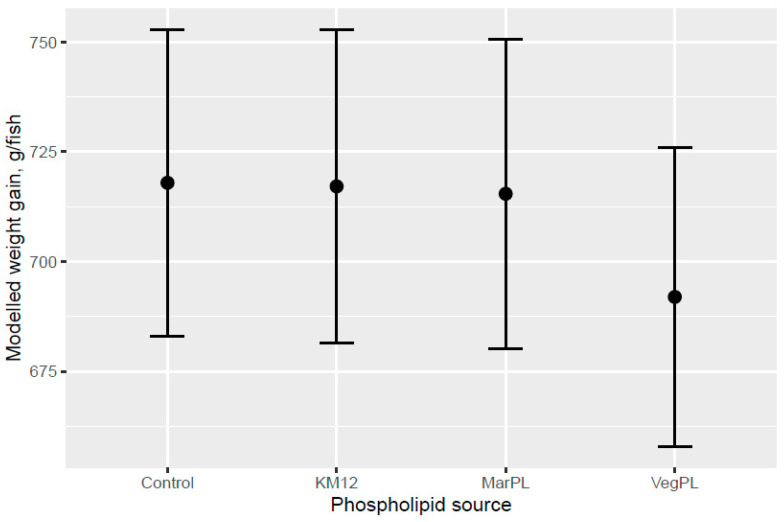
Modelled weight gain after feeding diets with different phospholipid sources for the whole trial, with 53 d in the freshwater pre-transfer phase on test diets followed by feeding the same commercial diet over 98 d in the common seawater tank. The bars are plotted with 95% credible intervals. Dots denote observed tank mean weight gains.

**Figure 7 animals-13-00835-f007:**
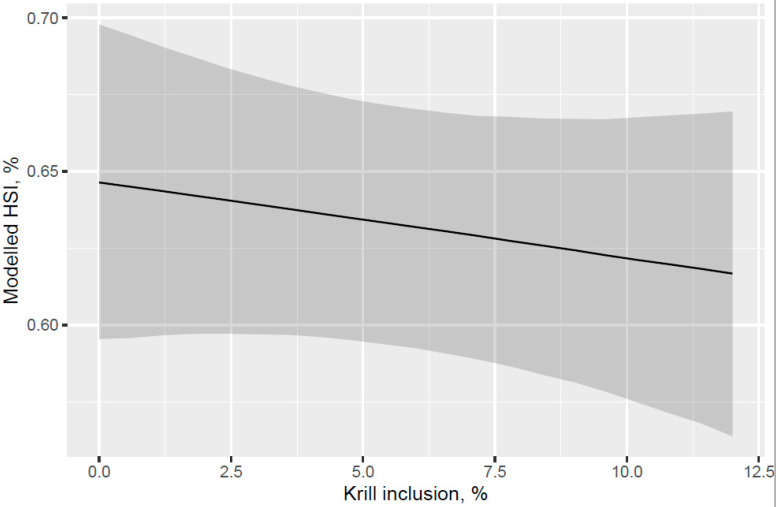
Modelled hepatosomatic index (HSI) with increasing KM inclusion after 53 d of feeding in the freshwater pre-transfer phase plotted with 95% credible intervals as the grey band. HSI is derived from the model for a 160 g average-sized sampled fish.

**Figure 8 animals-13-00835-f008:**
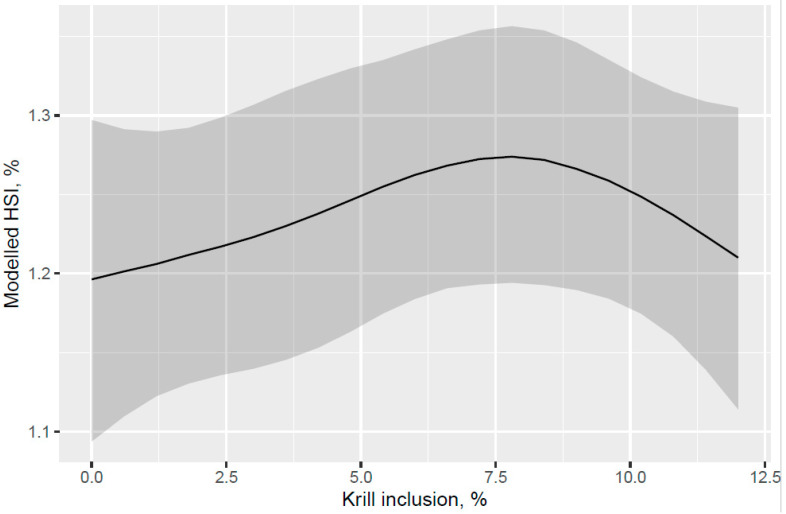
Modelled hepatosomatic index (HSI) for different phospholipid sources over 53 d in the freshwater pre-transfer phase followed by feeding the same commercial diet over 98 d in a common seawater tank plotted with 95% credible intervals as the grey band. HSI is derived from the model for a 792 g average-sized sampled fish.

**Figure 9 animals-13-00835-f009:**
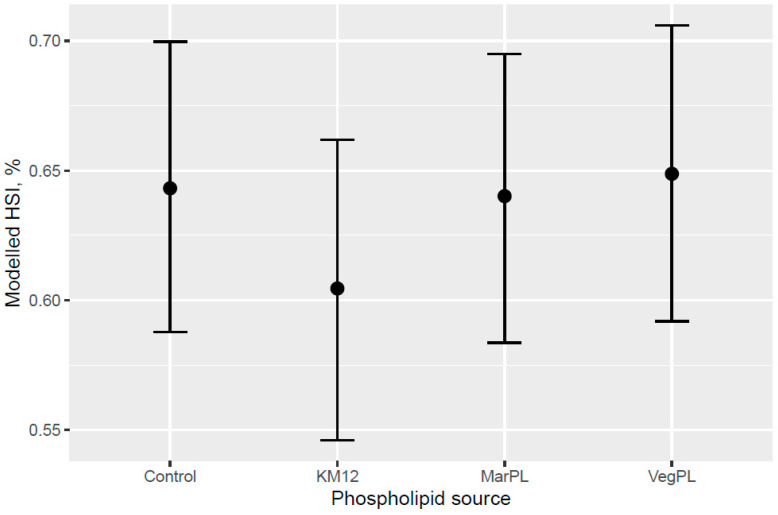
Modelled hepatosomatic index (HSI) for different phospholipid sources after 53 d of feeding in the freshwater pre-transfer phase plotted with 95% credible intervals. HSI is derived from the model for a 157 g average-sized sampled fish.

**Figure 10 animals-13-00835-f010:**
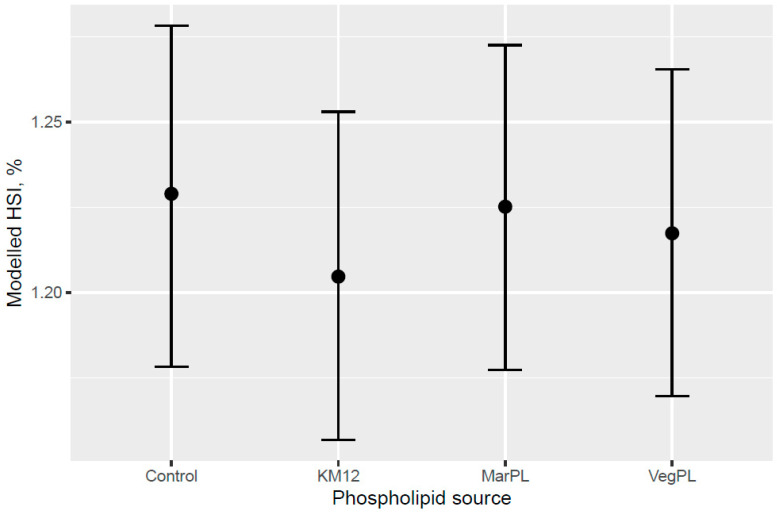
Modelled hepatosomatic index (HSI) for different phospholipid sources over 53 d in the freshwater pre-transfer phase followed by feeding the same commercial diet over 98 d in a common seawater tank plotted with 95% credible intervals. HSI is derived from the model for a 792 g average-sized sampled fish.

**Figure 11 animals-13-00835-f011:**
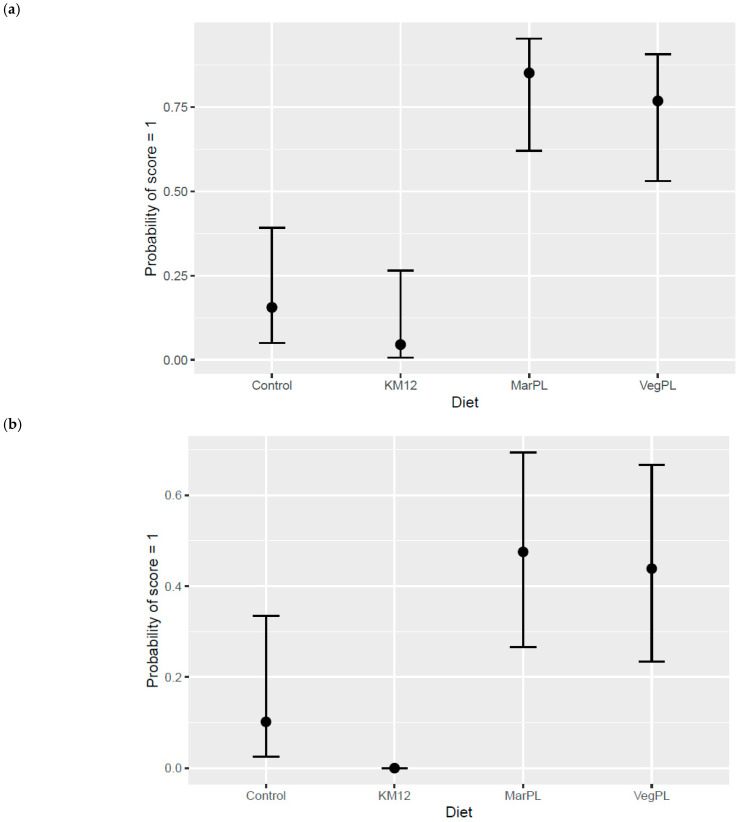
Modelled probability for gill histology score of 1 described as very mild to mild as the most severe observation over a 0 score (no lesions/normal tissue) for (**a**) inflammation of lamella and (**b**) hyperplasia in gills of the fish that were fed diets containing different phospholipid sources after 53 d of feeding in the freshwater pre-transfer phase. Error bars give 95% credible intervals.

**Table 1 animals-13-00835-t001:** Formulation and expected composition of test diets for the feeding trial in freshwater.

	Control	KM4	KM8	KM12	MARPL	VEGPL
Ingredient (% diet)						
Krill meal		4.0	8.0	12.0		
Marine PL					4.2	
Fluid soy lecithin						2.7
Fish oil	6.9	5.5	4.1	2.7	1.8	6.9
Plant oil	10.4	11.0	11.6	12.1	11.3	7.5
Fish meal (LT94)	10.0	10.0	10.0	10.0	10.0	10.0
Plant ingredients	68.8	65.9	62.9	59.9	68.8	68.6
Microingredients *	4.0	3.6	3.4	3.3	3.9	4.3
Total	100.0	100.0	100.0	100.0	100.0	100.0
Composition						
Protein (%; Dumas)	48.9	47.1	47.7	48.1	45.9	46.1
Fat (%; LfNMR)	24.0	24.4	24.3	24.2	22.7	21.8
Moisture (%; NIR)	6.0	6.5	6.1	5.9	6.3	8.2
Gross energy (MJ/kg; bomb calorimeter)	23.6	23.1	23.2	23.1|	22.1	22.7
Formulated PL% of diet from test ingredient	0	0.4	0.9	1.3	1.3	1.3

* Vitamins, minerals, and amino acids; * Formulated choline levels in excess of NRC 2011 requirements for salmonids (Section 2). Abbreviations: KM = krill meal; LfNMR = low-field nuclear magnetic resonance; NIR = near-infrared spectroscopy; PL = phospholipid.

## Data Availability

The data is available in the present manuscript and Appendix A.

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
