# Peer review of "Effects of Different Phospholipid Sources on Growth and Gill Health in Atlantic Salmon in Freshwater Pre-Transfer Phase"

_animals, 2023, doi:10.3390/ani13050835_

Round 1

Reviewer 1 Report (Previous Reviewer 1)

All the revision is carried out correctly.

Author Response

  1. Inclusion of all experimental diet in the abstract. 

    Authors have added the information in the revised manuscript, lines 23-26, which reads as follows:

    *There were 6 test diets in the FW phase: 3 diets with different doses of Krill meal (4, 8 and 12%), a diet with soy lecithin, a diet with marine PL (from fishmeal) and a control diet. Fish were fed a common commercial feed in the SW phase. *
  2. In section 2.2: What was time period to transfer from 0 ppt to seawater (Acclimation time?) 

    Authors have added the information in the revised manuscript, lines 150-156, which reads as follows:

    After this freshwater pre-transfer feeding period, fish from all the tanks (17-20 fish per tank from the 24 freshwater tanks) were transferred to a larger common seawater flow-through tank (5m diameter and 21.6m3 volume) with 28.5 ppt salinity, and no acclimation time from 0 ppt freshwater to 28.5 ppt seawater) with a water temperature drop at transfer (ca. 14 to 9℃) and crowding stress after transfer (lowered water level to ca. 20 cm for one hour with supplemental oxygen for all 459 fish of ca. 167g within a ca. 0 to 30h period after transfer)
  3. Is Chd- gene expression is studied? If not, than this section may be modified 

    Authors have modified the information in the revised manuscript, lines 397-400, which reads as follows:

    However, Chd expression was not studied in the current study, and hence, further studies are warranted to explore the association between Chd expression, HSI and absolute fat accumulation in liver in salmon.

Reviewer 2 Report (Previous Reviewer 2)

After adjustments made by the authors, the manuscript was improved and publication can be recommended.

Author Response

1. Inclusion of all experimental diet in the abstract. 

Authors have added the information in the revised manuscript, lines 23-26, which reads as follows:

There were 6 test diets in the FW phase: 3 diets with different doses of Krill meal (4, 8 and 12%), a diet with soy lecithin, a diet with marine PL (from fishmeal) and a control diet. Fish were fed a common commercial feed in the SW phase.

2. In section 2.2: What was time period to transfer from 0 ppt to seawater (Acclimation time?) 

Authors have added the information in the revised manuscript, lines 150-156, which reads as follows:

After this freshwater pre-transfer feeding period, fish from all the tanks (17-20 fish per tank from the 24 freshwater tanks) were transferred to a larger common seawater flow-through tank (5m diameter and 21.6m3 volume) with 28.5 ppt salinity, and no acclimation time from 0 ppt freshwater to 28.5 ppt seawater) with a water temperature drop at transfer (ca. 14 to 9℃) and crowding stress after transfer (lowered water level to ca. 20 cm for one hour with supplemental oxygen for all 459 fish of ca. 167g within a ca. 0 to 30h period after transfer)

3. Is Chd- gene expression is studied? If not, than this section may be modified. 

Authors have modified the information in the revised manuscript, lines 397-400, which reads as follows:

However, Chd expression was not studied in the current study, and hence, further studies are warranted to explore the association between Chd expression, HSI and absolute fat accumulation in liver in salmon.

This manuscript is a resubmission of an earlier submission. The following is a list of the peer review reports and author responses from that submission.

Round 1

Reviewer 1 Report

Dear Authors/ Editor,

Greetings!

Manuscript is well structured and informative. However, there are some correction and clarification, which need to be addressed before final acceptance.  

Abstract:

Mention all the six diet in the abstract section, which will make more clarity.

Value of HSI may be given.

MM: OK

2.2: What is crowding stress here?

What was time period to transfer from 0 ppt to seawater (Acclimation time?)

Salinity of seawater need to be mention in bracket

Statistical analysis: Unable to understand. Need to b reviewed by expert.

Results and discussion:

Gill histology: Highlighted part is not clear. Please rewrite. Similarly liver histology.

Carry over growth in marine environment is not noticed. Reason for this may be explained with suitable supporting references.

Is Chd- gene expression is studied? If not, than this section may be modiced.

Reviewer 2 Report

General comments 

The authors in the discussion of the manuscript report the issue of trend and there is no direct justification of the results. The results are not clearly explained in the discussion. The data need to be justified with the literature. The authors compare the data a lot with other works, but do not explain the results. There are no analyzes of the lipid profile of the products tested in fish diets. It is recommended to reject the manuscript

Introduction:

The authors need improved the objective of work, in the last paragraph in introduction. The authors need reply the question: What’s the objective of work? 

Material and methods 

Did the authors perform analyzes of the lipid profile of the diets? Why was it not carried out?

Is the amount of EPA and DHA present on the product label? Add values in the manuscript.

Add in the manuscript the description of the methodologies of analyzes of protein and fat in the diet.

Results and Discussion

The authors comment in the results and discussion that there are trends. Results should be either significant or not significant. Improve the results and discussion in this part of the manuscript.

“The present study evaluated different phospholipid sources fed over 53d in freshwater pre-transfer phase followed by feeding the same commercial diet over 98d in common seawater tank on growth performance and health parameters of Atlantic salmon. KM was evaluated in dose response (0.0, 4.0, 8.0 and 12.0% of diet), and diets with 2.7% fluid soy lecithin (VegPL) and 4.2% MarPL as alternative PL sources were formulated to provide the same added 1.3% PL of diet level as 12% KM. All the test diets contained 10% fish-288 meal in the FW phase.”  These part is results. Remove of manuscript. 

“Addition of KM did not give a clear increase in feed intake compared to the control diet and there was an indicated trend of decreased feed intake for MarPL and VegPL diets but cannot make strong conclusions due to the high variability”. The authors attributed “high variability” to what? the fishes? explain better.

“Feed intake could not be measured to isolate effects of pre-transfer diets due to fish being in a common tank and fed the same diet in the seawater phase” Why? very confused. 

“A trend for decreased hepatosomatic index (HSI) was indicated with increased KM” the decreased of HSI, is good for fish metabolism? 

KM provides astaxanthin (166 mg/kg in the KM used for the present study) to the diet as a natural antioxidant with potential anti-inflammatory properties (Fassett & Coombes, 2011). The authors measured astaxanthin in the diets? 

Conclusions 

Conclude with the most important results and in a direct way.

Reviewer 3 Report

The manuscript entitled “Effect of Different Phospholipid Sources on Growth and Gill Health in Atlantic Salmon in Freshwater Pre-transfer Phase” evaluates different sources of phospholipids in Atlantic Salmon before transfer to seawater phase. The topic is of interest for the salmon aquaculture industry and for the journal. However, many changes are necessary to the manuscript prior that it can be accepted for publication, especially statical analysis. Further information on the importance of phospholipids in salmon aquaculture should be provided. Moreover, it is necessary to mention the justification of using soy lecithin and marine phospholipid rich oil in the introduction section. The tile does not seem to contain the entire experiment. The title should include how the different sources of phospholipid affect not only the freshwater phase but also seawater phase.  

Specific comments:

Abstract

What was the hypothesis of the study? The hypothesis must be placed in the abstract and introduction sections.

ž  Line 21, 22, 28: Please provide standard error, such as 74 ± x g.

Introduction

ž  Line 36: Check format of all references and make them all consistent. As mentioned above, further information on the importance of phospholipids, justification of using soy lecithin and the other phospholipid and the hypothesis of the study should be provided. What is the relationship between phospholipids and crowding stress and how do phospholipids alleviate this stress? It seems to be the design of this experiment, but the explanation is insufficient.

ž  Line 59: Please provide references to the parameters evaluated in this experiment (liver or gill health, not the intestinal health).

ž  Line 77: Change “commercial 200 size diet” to “commercial feed”

Materials and Methods

ž  Line 78: The information in this section is quite incomplete. Author should include the analyzed values of proximate composition, energy content and fatty acid analysis of the diets. Proximate composition of diet must be performed after formulation and should provide analyzed values, not the formulated values.

ž  Create a section describing the proximate composition analysis process and explain the process (Moisture, Crude protein, fat, ash and energy).

ž  Where were the experimental diets made and what process did it go through? Please mention in this section.

ž  Line 97-103: This information is not required for this section. Move to the discussion section if necessary.

ž  Line 104 (Table 1): Please provide in detail which ingredients were used in the feed formulation. List all the ingredient used, not “plant ingredients”. Also, what type of plant oil was used? Provide all the specific details including micro ingredients. For example, how much vitamin premix, lysine, methionine or choline was added. As mentioned above, give the analyzed value of proximate composition, not the formulated value. Moreover, please clarify if proximate composition is based on dry matter or as fed and provide the sources of ingredients (where they are from).

ž  Line 110: Authors should include information regarding the type of system employed (flow through/RAS).

ž  Line 116: Elsewhere the initial fish size is stated to be 74g.

ž  Line 159: It is worth reconsidering the statistical method in the present study. What is the justification of choosing GAM over ANOVA? I recommend running stats with ANOVA for the freshwater phase (tank as an experimental unit) and with ANOVA or ANCOVA for the seawater phase (fish as an experimental unit). If ANOVA assumptions are not met (normality or variance issue), data transformation process needs to be carried out. Then I would recommend re-analyzing the results based on the significance.

Results

ž  Line 182: Results should be rewritten based on the different statistical method.

ž  Line 182: Please provide a table for growth performance and feed utilization including initial and final fish weight, weight gain, FCR, TGC and survival for freshwater phase and initial and final fish weight, weight gain and survival for seawater phase.

ž  Line 252: Please run other non-parametric analysis such as Kruskal-Wallis test followed by Wilcoxon’s post hoc test in order to draw conclusions based on significance.  

Discussion

ž  Line 282: The discussion section should also be rewritten based on the results obtained through the different statistical method.

ž  Line 285-289: This bit is repetitive from the Material and Methods, maybe can be deleted from the Discussion.

ž  Line 318: Although KO was not used in this experiment, too much unnecessary information is provided in the Discussion section on KO.

Figures: All the figures need to remade including significant levels in order to make a clear comparison between groups.